# Research on the classification of ship encounter scenarios based on CAE-LSTM

Taiyu Chai
School of Navigation
Wuhan University of Technology
Wuhan, China
282614@whut.edu.cn

Zhitao Yuan*
School of Navigation
Wuhan University of Technology
Wuhan, China
ztyuan@whut.edu.cn

Weiqiang Wang
School of Navigation
Wuhan University of Technology
Wuhan, China
weiqiangwang@whut.edu.cn

Shengjie Yang
School of Navigation
Wuhan University of Technology
Wuhan, China
yangshengjie@whut.edu.cn

*Abstract*— **To tackle the challenge of recognizing similar ship encounter scenarios under multi-ship interference coupling and dynamic evolution, this paper proposes a classification method that combines a Convolutional Auto-Encoder (CAE) and a Long Short-Term Memory (LSTM) recurrent neural network model. To extract many genuine ship encounter scenarios from historical AIS data for further categorization, first, a method for extracting ship encounter scenarios taking spatiotemporal proximity restrictions is devised. Then, by setting a time window and rasterizing the scenarios, a CAE-based model is constructed to characterize the spatial interference of ships in the scenarios. Further, an LSTM network is used to learn temporal evolution features, achieving a low-dimensional spatiotemporal vector representation of ship encounter scenarios. Finally, hierarchical clustering is applied to classify different ship encounter scenarios based on these low-dimensional spatiotemporal vectors. The proposed method is validated through extensive experiments using data from Ningbo-Zhoushan Port, and the results show that this method can effectively extract real ship encounter scenarios and accurately identify similar scenarios. This research provides robust support for a deep understanding of ship encounter scenarios and the mining of similar ship behavior patterns.**

*Keywords—ship encounter scenarios, scenarios classification, CAE, LSTM*

## I. INTRODUCTION

In recent years, the continuous growth in shipping volume has significantly increased maritime traffic density, leading to a rise in ship collision accidents [1]. Research shows that these mishaps are mostly caused by human factors. [2]. To mitigate collision incidents caused by human error, researchers have developed numerous navigation collision avoidance algorithms to enhance maritime safety[3]. Historical ship encounter scenarios contain rich avoidance processes and strategies. Extracting these scenarios and analyzing collision avoidance behavior patterns in similar situations allows this implicit knowledge to be integrated into the design of collision avoidance algorithms. This approach enhances the practicality of these algorithms and improves avoidance safety in similar scenarios. Therefore, extracting real ship encounter scenarios and effectively classifying similar scenarios hold significant potential for advancing collision avoidance algorithm design.

Ship encounter scenarios essentially involve interactions between multiple vessels, which can be explained through their trajectories. Because the Automatic Identification System (AIS) is widely used on ships, scholars can collect large quantities of high-quality vessel trajectory data at a low cost, providing a rich and reliable data source for extracting ship encounter scenarios. Related research on encounter scenario extraction using AIS data has been carried out by several academics. Through the use of AIS data, Ma Jie et al. [4,5] were able to successfully extract ship encounter scenarios by analyzing the spatiotemporal correlations during ship interactions. Similarly, Based on the spatiotemporal proximity relationships between ships, Wang et al.[6] identified ship encounter possibilities from AIS data, evaluated the significance of each event, and sampled the data to create test scenarios for collision avoidance algorithms.

Ship encounter scenarios are typical spatiotemporal sequence data, often exhibiting significant temporal evolution characteristics and complex multi-vessel interaction couplings. This complexity makes classifying ship encounter scenarios challenging. Current research mainly focuses on clustering analysis of individual ship trajectories. To identify frequent paths and discover abnormal trajectories, Li et al. [7] for instance, suggested a multi-step clustering methodology that combines principal component analysis, dynamic time warping, and an enhanced trajectory clustering center method. Ship itineraries were inferred from AIS data by Zhang et al. [8] using data-driven techniques such as ant colony optimization and geographic clustering of applications with noise based on density (DBSCAN). Zhang et al [9] classified ship trajectories using K-Means and DBSCAN clustering algorithms, then identified potential collision scenarios by detecting illegal evasive maneuvers through relative bearing angles and quantified the collision risk index when evasive actions were taken. However, these methods primarily rely on the similarity calculation of individual ship trajectories. Although they perform well in trajectory similarity analysis and classification, encounter scenarios involve the interactions of multiple ships, featuring significant temporal evolution characteristics and complex multi-ship interference effects. As a result, these methods have limitations in representing and measuring the spatio-temporal interference features in encounter scenarios and face challenges when directly applied to encounter scenario classification.

This paper is supported by the National Natural Science Foundation of China(NSFC) under Grant NO.52031009. (Corresponding author: Zhitao Yuan).

In recent years, deep learning has shown great potential in handling complex spatio-temporal data, and some studies have begun exploring its potential in trajectory similarity computation. These works demonstrate how deep learning techniques can more effectively capture the features of ship trajectories. Compared to traditional methods, deep learning models can automatically learn useful features from large amounts of data without relying on manual feature extraction, offering certain advantages [10]. Liang et al [11] proposed an unsupervised learning method based on a convolutional autoencoder (CAE), which maps trajectories into two-dimensional matrices to generate trajectory images and automatically extracts low-dimensional features via the CAE to compute similarity. Chen et al [12] introduced a method based on convolutional neural networks (CNN) to identify movement patterns in emerging trajectories. In this approach, a mobility-based trajectory structure is introduced as input to the identification model, and evaluations using real maritime trajectory datasets show the superiority of this method. Kontopoulos et al [13] proposed a novel method that integrates research in computer vision and trajectory classification, automatically extracting meaningful information from trajectory data and identifying movement patterns without the need for expert input.

Overall, unsupervised and semi-supervised methods based on deep learning are gradually gaining attention in the field of maritime situational awareness. These methods share a common feature: they reduce reliance on manual intervention through automatic feature extraction, demonstrating strong adaptability, especially when handling large amounts of unlabeled data. It is recommended to develop an unsupervised learning method for representing the complex temporal evolution characteristics of ship encounter scenarios to enable effective classification. Based on the above analysis, this study proposes a ship encounter scenario classification method that combines a Convolutional Autoencoder (CAE) with a Long Short-Term Memory (LSTM) network. This approach comprehensively considers both the spatial interference coupling features among multiple ships and the temporal evolution patterns within the encounter scenario, enabling effective classification of ship encounter scenarios.

## II. Methodology

This paper focuses on two main tasks: the extraction of real ship encounter scenarios based on AIS data, and the classification of these scenarios using a combination of CAE and LSTM models. As seen in Figure. 1., the research framework consists of three steps: preprocessing AIS data, ship encounter scenario extraction, and clustering ship encounter scenarios.

Step 1: Data Preprocessing. Original AIS data is preprocessed to retain key attributes such as timestamp, Maritime Mobile Service Identity (MMSI), ship length, longitude, latitude, speed over ground (SOG), and course over ground (COG). These attributes are essential for calculating the subsequent spatiotemporal relationships of the vessels.

Step 2: Encounter Scenario Extraction. Based on the spatiotemporal proximity analysis of ships, ship encounter scenarios are extracted from historical AIS data. This extraction provides numerous encounter scenarios that reflect the real navigational behaviors of ships for subsequent classification.

Step 3: Time Slicing and Gridding. Time slicing and gridding are applied to the scenarios to characterize their spatiotemporal attributes.

Step 4: Feature Representation. CAE and LSTM represent the spatial and temporal features of the encounter scenarios with feature vectors.

Step 5: Clustering of Encounter Scenarios. Hierarchical clustering is applied to the feature vectors of all scenarios. To achieve the classification of encounter scenarios, the ideal number of clusters is found using the Silhouette Coefficient (SC) index.

In summary, based on the most advanced research findings, our CAE-based ship encounter scenario classification method offers the following innovations. We propose generating information trajectory images by remapping the ship trajectories involved in encounter scenarios into two-dimensional matrices:

1. The similarity between different encounter scenarios is measured by assessing the structural similarity between the corresponding information trajectory images.

2. A convolutional autoencoder neural network is proposed to learn the low-dimensional representation of these images in an unsupervised manner. The learned representation can effectively capture the characteristics of ship encounter scenarios.

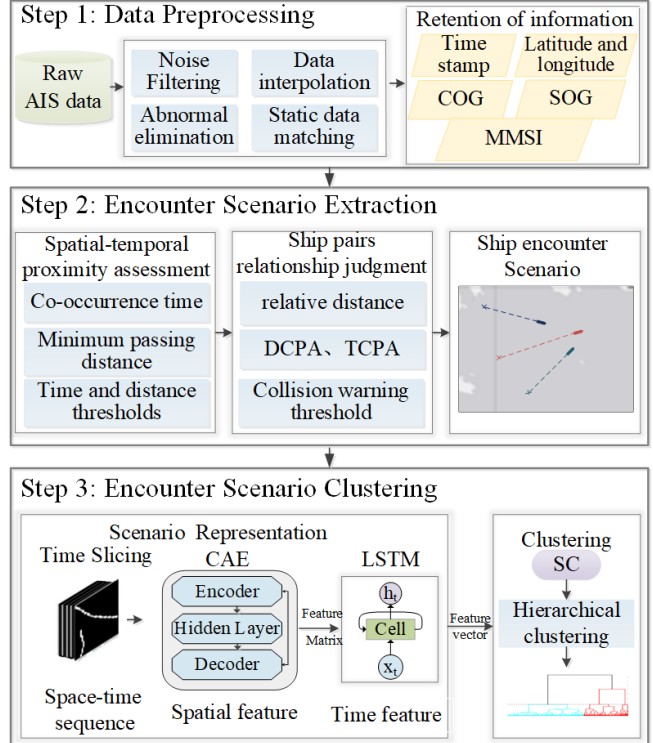

Fig. 1.  Overview of the proposed approach.

### A. Data Preprocessing

The quality of AIS data significantly impacts the accuracy of the extracted encounter scenarios. Due to various factors, AIS data may contain inconsistencies with the actual navigational

state of the ships. Therefore, preprocessing is necessary before extracting encounter scenarios[14]. Main preprocessing operations include noise filtering, anomaly removal, data interpolation, and matching of static data information[15].

### B. AIS Data-Based Encounter Scenarios Extracted

Spatio-temporal relationships between ships are fundamental for extracting encounter scenarios. In this work, ship encounter scenarios are described as a series of ship pairs, that within a specific time sequence, satisfy specific spatiotemporal proximity conditions. Figure 2. shows a graphical description of ship encounter scenarios. The timeline is shown on the x-axis in Figure 2, and the ship identification numbers that are part of the encounter scenarios are shown on the y-axis. The lines with arrows represent the navigation period of the Own Ship (OS) in the study area, while the lines with arrows in front of each Target Ship (TS) indicate the periods when the TS meets the preset spatiotemporal proximity conditions with the OS.

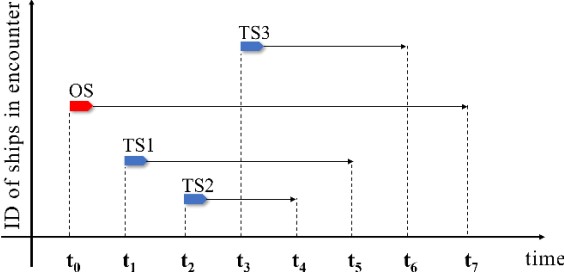

Fig. 2. Overview of the proposed approach.

Additional evolution analysis of the Distance at the Closest Point of Approach (DCPA) and the Time to the Closest Point of Approach (TCPA) is necessary to precisely define spatiotemporal proximity relationships between ships at each time[16]. By analyzing the preprocessed AIS data, the spatio-temporal relationships between ships can be extracted, allowing the identification of ship encounters. Specifically, when two ships remain in the study area for a period exceeding the set time threshold, the minimum distance between them is calculated. Further analysis will be done on their relative distance, DCPA, and TCPA evolution patterns if this closest passing distance is less than the distance criterion. A ship pair will be deemed to meet the spatiotemporal proximity constraints that may result in a collision if their relative distance is decreasing and stays within the early-warning distance, and both DCPA and TCPA values stay below a specific threshold before approaching the closest passing distance. Under such circumstances, the relevant data will be saved and the segments of two ships that satisfy these spatiotemporal proximity constraints will be retrieved. The beginning and ending times of the extracted segments, as well as static and dynamic information on each ship (such as MMSI, length, width, type, and so on) at each timestamp over this period, are all included in this data. Figure. 3. provides a graphical illustration of DCPA and TCPA, with the calculation formulas provided below.

$$DCPA_t = D_{ijt} \cdot \sqrt{1 - cos^2(\theta_{ijt})} \qquad (1)$$

$$TCPA_t = \frac{-D_{ijt} \cdot \cos(\theta_{ijt})}{v_{ijt}} \qquad (2)$$

where $D_{ijt}$ represents the distance between ship $i$ and ship $j$ at time $t$. $v_{ijt}$ represents the relative speed between ship $i$ and ship $j$ at time $t$. $\cos(\theta_{ijt})$ indicates the angle formed by the cosine of the relative velocity and the line joining the two ships.

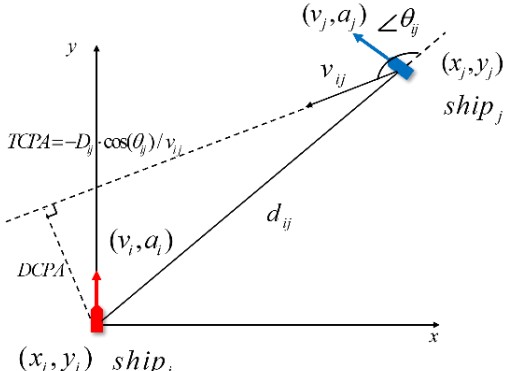

Fig. 3. DCPA and TCPA interpretation in graphics.

### C. Encounter Scenario time slice

Ship encounter scenarios, as spatiotemporal sequence data, involve mutual interference between ships that varies over time. Therefore, classifying encounter scenarios requires attention to both spatial interference characteristics and temporal evolution patterns of the ships. Time-slicing the scenarios and gridding each slice is the first step in the process of efficiently extracting the spatial and temporal features of these scenarios. This maps the temporal evolution of spatial interference characteristics into multi-time-window grids. Compared with the original trajectory image pixels, raster images contain richer information and are more conducive to CAE to characterize the interaction of ships in the encounter scenario.

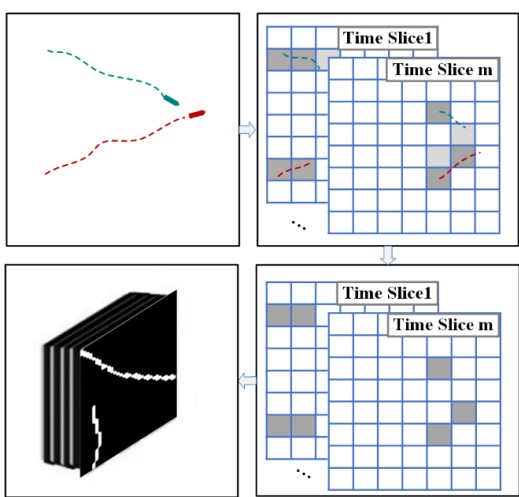

Fig. 4. Raster map generation and scene time slicing.

Thus, this paper projects the original ship trajectory into a two-dimensional matrix to generate a trajectory raster image

based on the time sequence of the encounter scenarios, maintaining the original spatiotemporal characteristics. To balance the information richness of encounter scene slices and the total number of slices, the time window duration is set to 3 minutes, and the time window step to 1 minute. The particular procedure is depicted in Figure. 4.

### D. Feature Representation of Encounter Scenarios

To fully representant spatial interaction features between ships from multi-time window raster images and to learn the contextual relationships between feature sequences, as well as to uncover the temporal evolution patterns of the scenarios, we employ a multi-layer CAE neural network combined with LSTM for unsupervised learning and feature representation. The CAE, with convolutional and pooling layers, learns to identify local spatial interactions and patterns within each raster image[17]. Once spatial features are obtained, they are fed into the LSTM model, which captures the temporal evolution of these features over multiple time windows. The combination of CAE and LSTM enables a comprehensive representation of both the spatial interactions between ships and their dynamic changes over time.

This study employs a CAE-based autoencoder architecture. Compared to traditional autoencoders, CAE incorporates convolutional and pooling layers, allowing for better extraction of local features related to ship spatial interference in the scene grid maps. As shown in Figure. 5, the CAE model consists of three convolutional layers, three max-pooling layers, and fully connected layers. The encoder layer transforms input scene grid maps into low-dimensional feature vectors, thereby representing the spatial features of encounter scenarios. The decoder layer uses ReLU as the activation function to effectively reconstruct the low-dimensional feature vectors into scene grid maps. Additionally, to enhance the feature representation capability CAE, this study introduces a loss function sensitive to the structure of the images, specifically the structural similarity (SSIM) index, to ensure the accuracy of the extracted features. To further elucidate the working mechanism of the CAE model, the operations of convolutional and fully connected layers are described in detail as follows:

$$x_k^l = A_E(f_k^l \odot x_k^{(l-1)} + b_k^l) \quad (3)$$

$$Y = \mathcal{H}(x) = wx + \beta \quad (4)$$

where $l$ represents the layer number, $\odot$ denotes the convolution operation, $f_k^l$ represents the convolution kernel, $x_k^{l-1}$ represents the feature map, $b_k^l$ is the bias term, and $Y$ is the feature vector with a final output dimension $L$. The loss function, through training the model, ensures that the reconstruction $\tilde{x}$ of the decoder output has minimal error relative to the original input $x$. The following is the definition of the loss function SSIM:

$$\mathcal{F}(x, \tilde{x}) = 1 - \frac{1}{M} \sum_{m=1}^{M} \text{SSIM}(x, \tilde{x}_m) \quad (5)$$

$$\text{SSIM}(x_m, \tilde{x}_m) =$$
$$\frac{2\mu_{x_m}\mu_{\tilde{x}_m} + c_1}{\mu_{x_m}^2 + \mu_{\tilde{x}_m}^2 + c_1^2 \sigma_{x_m}^2 + \sigma_{\tilde{x}_m}^2 + c_2} \quad (6)$$

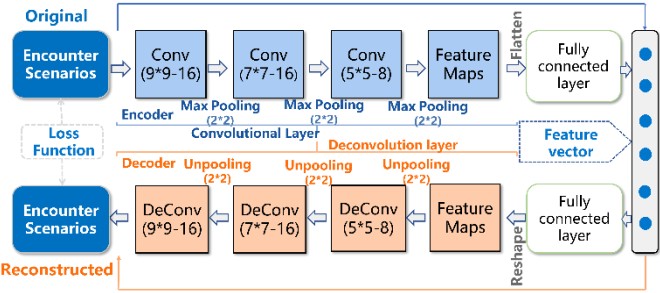

Fig. 5. The architecture of convolutional autoencoder.

LSTM is widely used for studying persistent features in time series data and can effectively learn dependencies between time series[18]. Therefore, LSTM is chosen to represent temporal feature evolution. The LSTM primarily consists of three gating units: the forget gate, the input gate, and the output gate, as shown in Figure. 6. The forget gate controls the transmission or forgetting of information. The process is described by Equation (7):

$$f_t = \sigma(W_f \cdot [h_{t-1}, x_t] + b_f) \quad (7)$$

where $W$ represents weight, $b$ represents bias, $[h_{t-1}, x_t]$ represents a vector consisting of the hidden layer output $h_{t-1}$ of the previous LSTM module, and the input $x_t$ of the current module, $\sigma(\cdot)$ represents the sigmoid function.

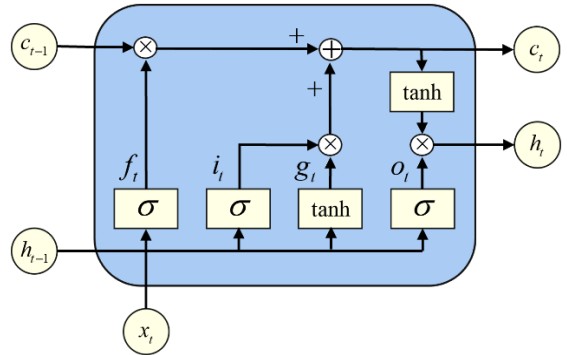

Fig. 6. LSTM unit structure diagram.

### E. Clustering of Encounter Scenarios

Feature vectors can eventually describe the intricate spatial relationships and temporal evolution of ship encounter events through the aforementioned method. By calculating the distance between each related feature vector, the similarity between ship encounter scenarios is determined. Once the distances are obtained, clustering algorithms classify the scenarios, and the results are evaluated using metrics to obtain the final classification outcome. Hierarchical clustering, simple and widely used, can reflect the step-by-step partitioning process of each object through a hierarchical clustering tree.

[19,20]Therefore, hierarchical clustering is chosen as the clustering algorithm for this study's encounter scenarios.

In the process of hierarchical clustering, it is challenging to directly select the best clustering result Therefore, an indicator is needed to select the appropriate number of clusters. In this paper, the value of $k$ is adaptively determined using the silhouette coefficient. $SC$ is defined by the mean distance from any point in the cluster to other points in the cluster after classification and the mean distance from any point to all points in the adjacent clusters. The better the categorization effect, the higher the SC value. The formula (8) displays the $SC$ calculating procedure.

$$SC(i) = \frac{CTb(i) - CTa(i)}{\max\{CTa(i), CTb(i)\}} \tag{8}$$

The average distance between scenario $i$ and other scenes in the same cluster is $CTb(i)$, whereas the minimal average distance between scenario $i$ and other clusters is $CTa(i)$. The silhouette coefficient ranges from -1 to 1, with higher values indicating better clustering performance.

## III. CASE STUDY

### A. Data collection and processing

This research uses data from November 1, 2018, to November 30, 2018, for the outside waters of Ningbo-Zhoushan Port. As shown in Fig.7, the targeted area is situated between latitudes 29°30N-29°49 N and longitudes 122°20E-122°60 E. To guarantee the precision of the ship encounter scenario analysis, specific mission vessel data, including tugboats, fishing boats, and anchored ships, were removed from the data. Subsequently, the residual data underwent data preprocessing procedures in preparation for more experiments. It is evident from the trajectory distribution that there are a lot of ship interactions in the research area.

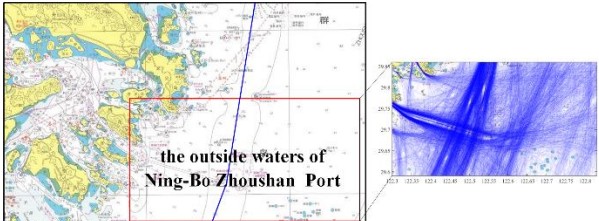

Fig. 7. The location of the study area.

### B. Analysis and validation of scenario extraction results

Three sample ship encounter scenarios are shown in Figure 8 to confirm the retrieved ship encounter scenarios. Four graphics are used to explain each scenario: the first graph shows the encounter process from start to finish using printed trajectories. The end state of the interaction is indicated by the ship icon in this subgraph. The progression of relative distance, DCPA, and TCPA between the OS and other TSs during the encounter process is shown in the remaining three graphs (a), (b), and (c). In these cases, the DCPA stays tiny for a while, the TCPA changes from positive to negative, and the relative

distance first drops to a very low value before gradually increasing. The retrieved scenarios are validated by the evolutionary patterns that align with real-world encounter experiences. The aforementioned evolution trends of relative distance, DCPA, and TCPA are all consistent across all extracted situations.

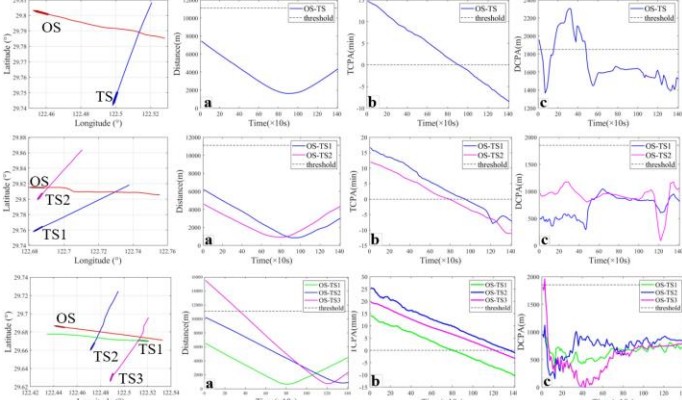

Fig. 8. Encounter situations involving varying numbers of ships and the development of their features.

Due to computational cost constraints, experimenting with all ship encounter scenarios is difficult. Therefore, selecting common encounter scenarios in maritime navigation as experimental data is necessary. As seen in Figure 9, the extracted encounter scenarios were first categorized and statistically examined according to the number of ships engaged. According to the classification results, two-ship encounters make up around half of all extracted scenarios, making them the most frequent. As ships involved increase, the number of scenarios gradually decreases, with a substantial decline occurring when the number of ships exceeds five.

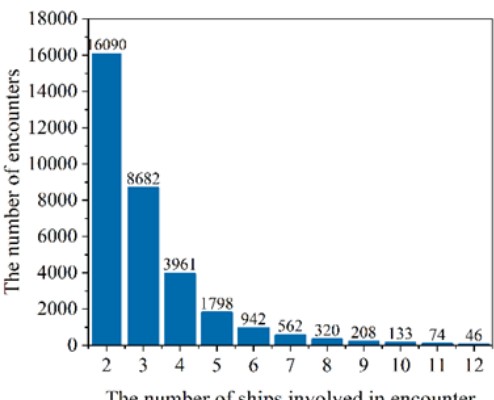

Fig. 9. Scenario classification outcomes depending on the number of ships.

To ensure the experimental data is representative while also saving computational costs, two-ship and three-ship encounter scenarios are chosen as the experimental dataset. This selection includes common two-ship encounters and more complex multi-ship encounters, which occur more frequently in actual maritime navigation. The durations of the two types of encounter scenarios in the experimental dataset were then statistically analyzed, and Figure. 10. displays the results. The analysis revealed that the proportions of two-ship and three-ship

scenarios lasting more than 10 minutes were 84.6% and 90.1%, respectively. This data segment is representative of all data exceeding 10 minutes, providing an important reference value for experimental analysis. Based on maritime navigation experience, scenarios lasting 10-20 minutes were chosen as experimental data. This selection ensures the significance of ship interactions while preventing the dataset from becoming overly large. Therefore, two-ship and three-ship encounter scenarios lasting 10-20 minutes were chosen as the final experimental dataset.

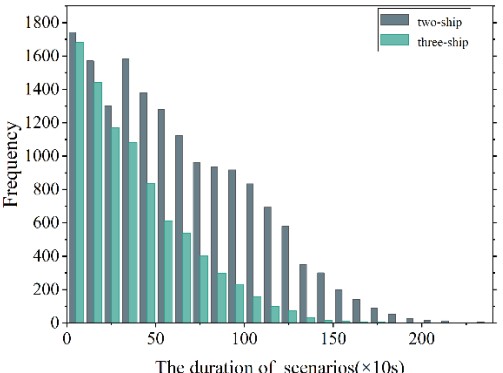

Fig. 10. Duration statistics for encounter scenarios.

## C. Experimental Software Environment and Model Training

For the experimental software environment, Python was chosen, using the PyTorch deep learning framework to train the model. The hyperparameter settings are shown in Table I. In Table I, Adam is the optimizer for the adaptive moment estimation method; Batch size represents the number of samples trained in each batch; Epoch refers to the number of training epochs; and Num Hidden Unit is the hidden layer dimensions of the LSTM.

TABLE I. HYPERPARAMETER SETTINGS

| HYPERPARAMETER | Parameter Value |
|---|---|
| Optimizer | Adam |
| CAE hidden layer dimensions | 8 |
| Batch size | 128 |
| Learning Rate | 0.001 |
| Epoch | 760 |
| Num Hidden Unit | 3 |

A total of 500 scenarios were selected from the experimental dataset for model training. First, the encounter scenarios were time-sliced, resulting in 7,366 and 7,261 scenario grid images, respectively. These encounter scenarios were then input into the CAE to extract spatial features. After 760 training epochs, the change in the loss function values with the number of training epochs is shown in Figure. 11. The training error converges to a very small value, indicating that the trained CAE can reconstruct the input data from the latent layer features. To demonstrate that the trained CAE can reconstruct the original encounter scenarios, the original scenario images and their reconstructed versions are shown in Figure. 12. The first row displays the original ship

encounter scenarios, while the second row shows the reconstructed images. The structural similarity between the original and reconstructed scenarios demonstrates that the CAE model excels in capturing low-dimensional representations and reconstructing high-quality images from these features. Finally, the feature matrix generated by the CAE is input into the LSTM model to learn the spatial feature evolution of the scenarios over time, outputting feature vectors to represent them.

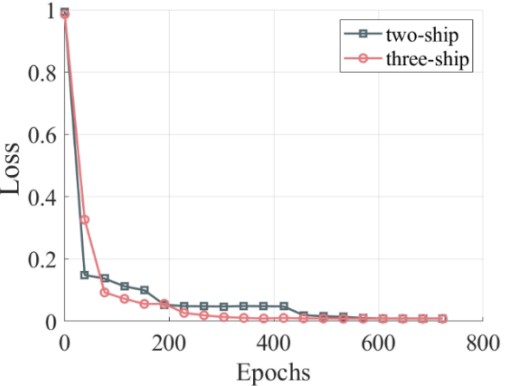

Fig. 11. Loss during the training of CAE.

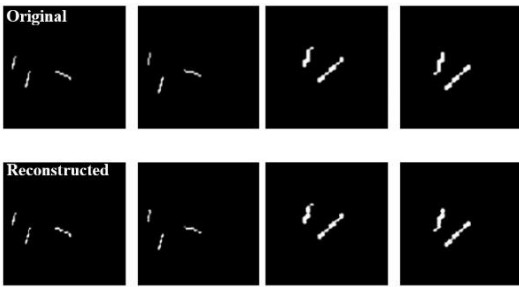

Fig. 12. Original and reconstructed encounter scenario images of CAE

## D. Clustering and Evaluation

The ship encounter scenarios were represented by feature vectors using the CAE-LSTM approach. Subsequently, hierarchical clustering was applied to these feature vectors to classify the ship encounter scenarios and obtain clustering results. SC was used to determine the ideal number of clusters and evaluate the effectiveness of clustering. Cluster counts varied from two to fifteen., and the *SC* values varied accordingly, as shown in Figure. 13.

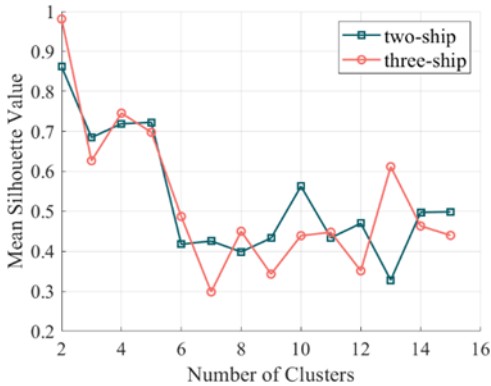

Fig. 13. Variation of silhouette coefficient values with the number of clusters

It demonstrates that both datasets obtained the highest silhouette coefficient values when there are two clusters. However, avoiding too few clusters is required to ensure a detailed separation of the microscopic aspects of ship interactions in various encounter scenarios. Therefore, 5 and 4 were chosen as the final number of clusters for the two datasets, respectively. These values represent the inflection points of the silhouette coefficient for both datasets. Beyond these points, as the number of clusters increases, the silhouette coefficient generally declines, indicating a deterioration in clustering performance.

After clustering the encounter scenarios, the frequency and duration distributions for each cluster are shown in Figures 14 and Figure. 15, respectively. For further analysis, the two clusters with the highest and lowest frequencies from each dataset were selected for feature analysis.

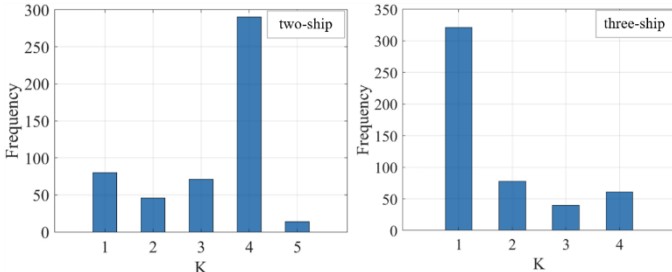

Fig. 14. Frequency distribution of encounter scenarios.

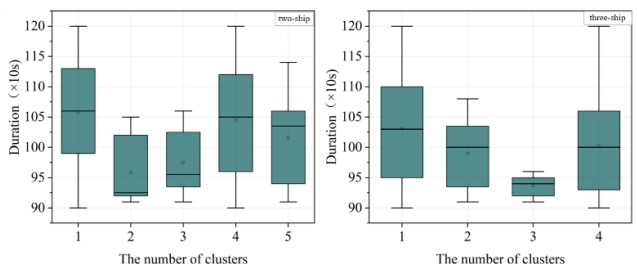

Fig. 15. The duration distribution of each cluster of encounter scenarios.

The interaction process between ship trajectories and the evolution of two features—relative distance and TCPA is shown in Figures 16 and Figure. 17. The first row of three images shows the complete trajectory of three encounter scenarios, where "○" and "×" represent the start and end positions of the encounter scenario, respectively. The relevant scenarios' relative distance and TCPA evolution are shown in the other two rows. The first two columns belong to the same cluster and illustrate the common characteristics of the scenarios. The third column represents a different cluster to highlight the distinctions.

For the two-ship encounter scenarios, Cluster 4 features ships moving in opposite directions, showing a head-on encounter with the relative distance initially decreasing and then increasing, and the TCPA exhibiting a linear decreasing trend. Cluster 5, on the other hand, consists of ships moving in the same direction, with the relative distance remaining constant and TCPA showing a decreasing trend but with significant fluctuations. For the three-ship encounter scenarios, Cluster 1 involves one target ship crossing paths with the OS, while the other target ship encounters head-on. The relative distances for

both target ships initially decrease and then increase, with the increase varying in magnitude. The TCPA shows a decreasing trend, with one ship's TCPA decreasing linearly and the other exhibiting noticeable fluctuations. In contrast, Cluster 3 features both target ships crossing paths with the OS. Although the relative distance trend is similar to Cluster 1, the ships in Cluster 3 are moving in the same direction, resulting in consistent changes in relative distance and TCPA fluctuating consistently before reaching zero. In summary, the interaction of trajectories, the evolution of features, and the duration within the same cluster exhibit consistent patterns. Different clusters, however, show distinctly different patterns.

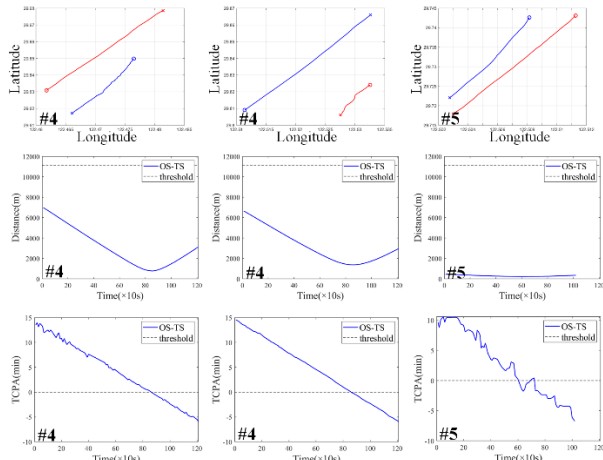

Fig. 16. Trajectory interaction and feature evolution process of the two-ship encounter scenarios.

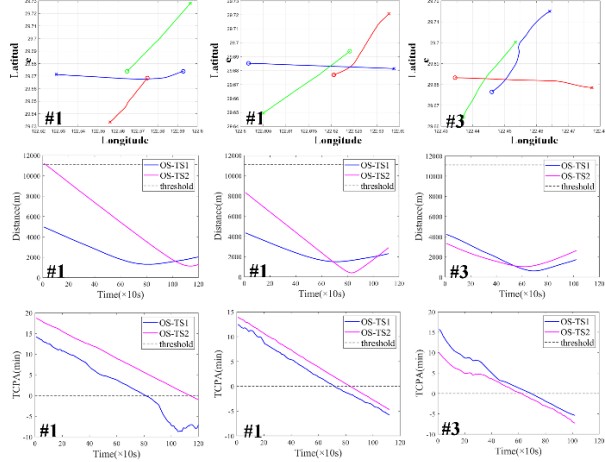

Fig. 17. Trajectory interaction and feature evolution process of the three-ship encounter scenarios.

Through the above analysis, the ship encounter scenario clustering method proposed in this paper effectively classifies different scenarios. The visual verification of trajectory interactions and feature evolution during the encounter process confirms the validity of this classification method. It demonstrates the various interaction patterns and contexts among multiple ships in complex navigable waters, aiding in distinguishing and understanding different types of ship encounter scenarios.

## IV. CONCLUSION

This paper proposes a method for clarifying ship encounter scenarios. First, ship encounter scenarios are segmented using time windows, and convolutional autoencoders generate spatial feature vectors for each time slice. Next, these spatial feature vectors are sequentially input into a long short-term memory (LSTM) network to produce temporal feature vectors. Finally, hierarchical clustering is applied to group the feature vectors based on their spatiotemporal attributes. Experimental results demonstrate that this method effectively classifies encounter scenarios involving various numbers of ships. The visualization of the interaction process and the dynamic evolution of features between ships confirms the classification's effectiveness.

## V. FUTURE WORK

In the future, we plan to make improvements in the following two directions:

1. Increase the size of the experimental data sample and optimize the scenario construction method to develop a multi-ship encounter scenario library tailored for complex navigational waters. Additionally, establish a query index based on ship scenarios.

2. Improve the classification method of ship encounter scenarios and enrich the dynamic characterization of encounter scenarios; design relevant application algorithms based on the scenario library, such as scenario prediction, risk assessment, and ship collision avoidance algorithms, etc., and further study the characterization of multi-ship encounter scenarios and the evolution law in depth.

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
