# OpenReview forum: "Research on the classification of ship encounter scenarios based on CAE-LSTM"
_IEEE.org/ICIST/2024/Conference — IEEE ICIST 2024 Conference Submission_

### Official Review · Reviewer_r7e2 · 2024-08-21
**accept**

**Rating:** 7
**Confidence:** 3

**Review:**

Comment:
This study proposes a method for clarifying ship encounter scenarios. The theory is correct and can be accepted after responding the following comments.
(1)	The contribution of the article needs to be written in the abstract.
(2)	There are many typos and grammar errors. The authors should have a native English speaker or software packages to perform the editing check.
(3)	The conclusion of the article suggests using the present perfect tense for description.

---

### Official Review · Reviewer_o8WG · 2024-08-22
**this work is well organized and appears potentially interesting, it can be accepted with a little modification.**

**Rating:** 8
**Confidence:** 4

**Review:**

This paper proposes a method for clarifying ship encounter scenarios. First, ship encounter scenarios are segmented using time windows, and convolutional autoencoders generate spatial feature vectors for each time slice. Next, these spatial feature vectors are sequentially input into a long short-term memory network to produce temporal feature vectors. Finally, hierarchical clustering is applied to group the feature vectors based on their spatiotemporal attributes. In general, this work is well organized and appears potentially interesting, it can be accepted with a little modification.
1.	To ensure the visual quality is optimal, it is recommended that the author provide higher resolution images to clearly showcase the relevant details and information.
2.	What are the future research directions outlined in this article?
3.	What are the innovative aspects of this system compared to others?
4.	Please explain the derivation of Equation (8).

---

### Official Review · Reviewer_HGnw · 2024-08-28
**This article is very interesting and a good one**

**Rating:** 7
**Confidence:** 3

**Review:**

This paper proposed a classification method that combines a Convolutional Auto-Encoder (CAE) and a Long  Short-Term Memory (LSTM) recurrent neural network model. The obtained result is valuable and can be accepted if the following problems can be clarified.
(1)	In the introduction, the shortages of those relevant studies are suggested to be further summarized.
(2)	There exist several spelling and grammar errors. Please check carefully and further polish.
(3)	In the Clustering and Evaluation section, “Figures. 16. and Figures. 17.” need to be modified to “Figures 16 and Figure. 17.” Please strictly check for similar errors and correct them.
(4)	The references require updating and standardization of their format to ensure consistency and accuracy.

---

### Decision · Program_Chairs · 2024-09-06

Accept (Oral)